# Robustness Analysis of an Urban Public Traffic Network Based on a Multi-Subnet Composite Complex Network Model

**DOI:** 10.3390/e25101377

**Published:** 2023-09-25

**Authors:** Gengxin Sun

**Affiliations:** College of Computer Science & Technology, Qingdao University, Qingdao 266071, China; sungengxin@qdu.edu.cn

**Keywords:** urban public traffic network, multi-subnet composite complex network, cascading failure model, robustness, high-order complex network

## Abstract

An urban public traffic network is a typical high-order complex network. There are multiple types of transportation in an urban public traffic network, and each type has different impacts on urban transportation. Robustness analyses of urban public traffic networks contribute to the safe maintenance and operation of urban traffic systems. In this paper, a new cascading failure model for urban public traffic networks is constructed based on a multi-subnet composite complex network model. In order to better simulate the actual traffic flow in the composite network, the concept of traffic function is proposed in the model. Considering the different effects of various relationships on nodes in the composite network, the traditional cascading failure model has been improved and a deliberate attack strategy and a random attack strategy have been adopted to study the robustness of the composite network. In the experiment, the urban bus–subway composite network in Qingdao, China, was used as an example for simulation. The experimental results showed that under two attack strategies, the network robustness did not increase with the increase in capacity, and the proportion of multiple relationships had a significant impact on the network robustness.

## 1. Introduction

With the development of the economy and technology, as well as the increase in urban population, the connection between different types of urban public transportation vehicles has become closer [1,2]. A subway rail transit sub-network and a bus transportation sub-network jointly form an urban public transportation composite network, which not only facilitates daily life, but also has great hidden dangers. In the case of natural disasters, man-made damage and other emergencies [3,4], road congestion and crowds easily occur, which then affect the traffic of other road sections, or even cause a large area of urban traffic network paralysis, which is called the cascading failure phenomenon of urban traffic networks [5]. Research on the robustness of urban transportation networks has great practical significance for maintaining the normal operation of urban transportation networks.

Many researchers [6,7,8] believe that the traditional research on network congestion and network transmission capabilities mostly focuses on single-layer network structures. Ferber et al. [9] studied the robustness of the urban rail transit networks in London and Paris under deliberate and random attacks, and the simulation results indicated that the network would collapse if 0.5% of the stations were attacked continuously. Based on the edge weight function, Huang et al. [10] proposed a traffic redistribution strategy and constructed a cascading failure model for urban rail transit networks, which indicated that by appropriately increasing the number of key nodes and the capacity of easily congested nodes, as well as reducing the number of congested nodes in the initial stage of cascading failure, a redistribution strategy can effectively narrow the scope of cascading failure. This cascading failure research is based on single networks. However, in the real world, some complex systems can no longer be described by a single network structure; they are interdependent or coupled to form multi-layer networks. For example, an actual urban traffic network contains a variety of traffic networks. Combining multiple networks to form a high-order complex network and studying the robustness of the high-order complex network under cascading failure have become hot research topics. Buldyrev et al. [11] first investigated the cascading failure of two-layer interdependent networks and found that the cascading failure of coupled networks was different to that of single networks, which laid the foundation for subsequent research on the cascading failure of interdependent networks. Cai et al. [12] proposed a multi-layer coupled network cascading failure model based on an analysis of inter-layer dependency relationships, and believed that too high or too low inter-layer tightness would cause cascading failure to expand. Aleta et al. [13] proposed a double-layer urban public transport network model. Based on this model, they studied the traffic network structure of nine cities and proved that the proposed model can truly reproduce an urban traffic network. They also proved that the representation of each line and each type of transportation is useful for assessing the vulnerability of transportation systems. Shen et al. [14] constructed a two-layer network model of public transportation and subways, adopting a load redistribution strategy of local load loading in the direction of nodes. When the sub-network is attacked, a coupling effect between the networks can effectively alleviate the cascading failure phenomenon. Li [15] established a multi-layer network model including aviation networks, railway networks, waterway networks and public transport networks, and combined it with the transmission mechanism of infectious diseases to study the spread of COVID-19 in China. The results showed that the model had good robustness and reliability, and the multi-layer network model can be applied in many fields.

The above research did not consider the complexity of node attributes in the real world, and the different general meanings of nodes are difficult to represent in lower-order network models. Wang et al. [16] used a high-order complex network model to analyze the robustness of a bus–subway composite network and found that the composite network is better than the single network. Sun et al. [17] considered the actual traffic flow and adopted a residual capacity allocation strategy to construct a cascading failure model for composite transportation networks. Under deliberate attacks, the simulation results showed that the network’s survivability decreased by 90%, and there was a threshold for overload capacity parameters.

In summary, the existing research on cascading failures of urban transportation networks is mostly based on a single network or a hierarchical network model, and the fact that different transportation types in the actual transportation network are preferred by passengers for different reasons such as maintenance costs, operating speeds and other reasons has not been taken into account. In this paper, based on a multi-subnet composite complex network model, the impact of different proportions of multiple relationships on the node load in an urban public traffic network is considered, and based on the actual traffic flow in the network, a traffic function is set to improve the traditional load capacity model. Furthermore, a cascading failure model for urban transportation composite networks is constructed. Finally, the public transportation network in Qingdao is taken as an example for simulation, a public transportation composite network is constructed and the robustness characteristics of the composite network are analyzed under deliberate attacks and random attacks.

The contributions of the research are as follows:(1)As a result of the difference in the traffic flow of each line and each station, a new flow function algorithm is proposed.(2)The traditional load capacity model is improved through different types of transportation and multiple attributes of nodes.(3)Based on a high-order complex network model, the degree centrality algorithm is improved to select important nodes for deliberate attack simulation experiments.

## 2. Materials and Methods

### 2.1. Multi-Subnet Composite Complex Network Model

A multi-subnet composite complex network model [18] is a typical high-order complex network model. It can be defined as the combination of four elements: G=(V,E,R,F).(1)$V=\left\{v_{1},v_{2},\ldots ,v_{n}\right\}$ represents a set of nodes, n is the total number of nodes in the network;(2)$E=\{<v_{h},v_{l}>|v_{h},v_{l}\in V,1\leq h\leq n,1\leq l\leq n\}\subseteq V\times V$ represents a set of edges between nodes;(3)$R=R_{1}\times \cdots \times R_{i}\cdots \times R_{s}=\{(r_{1},\cdots r_{i},\cdots r_{s})|r_{i}\in R_{i},1\leq i\leq s\}$ represents a set of various relationships between nodes, $R_{i}$ represents the i-th relationship between nodes, s is the total number of relationships;(4)*F: E* → *R* is a mapping function.

An example of a composite network is shown in Figure 1.

In Figure 1, nodes represent different individuals and symbols on the edges represent various relationships between nodes. In the example network, $R=R_{1}\times R_{2}\times R_{3}$, $R_{1}=\{r_{1}\}$, $R_{2}=\{r_{2}\}$, $R_{3}=\{r_{3}\}$. There is only one relationship on edges $<v_{1},v_{2}>$,$<v_{2},v_{3}>$ and $<v_{4},v_{5}>$, and there are multiple relationships on edges $<v_{1},v_{3}>$,$<v_{2},v_{5}>$,$<v_{5},v_{6}>$,$<v_{5},v_{7}>$. Therefore, nodes $v_{1}$, $v_{2}$, $v_{3}$, $v_{5}$, $v_{6}$ and $v_{7}$ are affected by multiple relationships.

### 2.2. Cascade Failure Model Based on an Urban Public Traffic Composite Network

The cascading failure model includes the definition of the node initial load, the node capacity, and the load redistribution strategy. In traditional cascading failure models, nodes with higher degrees often hold higher loads, and the allocation of initial loads is also influenced by the degree of neighboring nodes. The initial load of the nodes is defined as:(1)$${L_{v_{i}}^{0}=(k_{v_{i}}\sum_{v_{m}\in \Gamma_{v_{i}}}{\;k}_{v_{m}})}^{\alpha },i=1,\;2,\ldots N$$
where $L_{v_{i}}^{0}$ represents the initial load of node $v_{i}$, $k_{v_{i}}$ represents the degree of node $v_{i}$, $\Gamma_{v_{i}}$ represents the neighboring nodes of node $v_{i}$, α is an adjustable parameter and N is the total number of nodes in the network.

The node capacity is the maximum load that a node can hold, and the larger the capacity, the less likely a node is to fail. The node capacity is positively correlated with the initial load and is defined as:(2)$$C_{v_{i}}=(1+c)\;L_{v_{i}}^{0}$$
where $C_{v_{i}}$ represents the node capacity of node $v_{i}$ and c is an adjustable parameter.

When a node fails, load redistribution is required to allocate its load to neighboring nodes that have not failed according to a certain strategy.
(3)$$\Delta L_{v_{j}v_{i}}=L_{v_{i}}\frac{L_{v_{j}}^{0}}{\sum_{v_{k}\in \Gamma_{v_{i}}}L_{v_{k}}^{0}}$$
where $\Delta L_{v_{j}v_{i}}$ represents the load which is allocated by failed node $v_{i}$ to its neighbor node $v_{j}$, $L_{v_{j}}^{0}$ represents the initial load of node $v_{j}$ and $L_{v_{i}}$ represents the load of node $v_{i}$.

When the sum of the initial load of node $v_{j}$ and the redistributed load exceeds its own capacity, node $v_{j}$ will also fail. This process is defined as follows:(4)$$L_{v_{j}}^{0}+\Delta L_{v_{j}v_{i}}>C_{v_{j}}$$

When node $v_{j}$ fails, load redistribution will cause other nodes to fail, which would result in cascading failure. When no nodes fail, load redistribution will no longer be carried out, which means the cascading failure process ends.

The traditional cascading failure model is only suitable for the cascading failure phenomenon in a single-relationship network, while actual urban public transportation systems contain a variety of transportation types, such as buses and subways. In this paper, a multi-relationship composite public transportation network is constructed; the relationships on the edges represent various transportation types, traffic flow acts as the load and the load is spread through the edges. The traditional cascading failure model is improved from three aspects:(1)Initial load

The establishment of stations in urban public transportation networks is usually related to the traffic flow near the stations. The higher the traffic flow, the more bus and subway lines. When two adjacent stations are in the same subnet (subway subnet or bus subnet), there is only one relationship between nodes. When the two stations are not in the same subnet, there are multiple relationships between nodes. Therefore, the traffic flow at each station is different, the traffic flow on each line is also different and the traffic flow of different modes of transportation also varies. In a composite network, the traffic flow of the edges is related to the relationship degree between nodes. The flow on edge $<v_{w},v_{h}>$ can be defined as:(5)$$flow(<v_{w},\;v_{h}>)=\sum_{\begin{array}{c} r_{j}\in R\\ r_{k}\in R \end{array}}\frac{k_{v_{w}}^{r_{j}}}{N_{j}}+\frac{k_{v_{h}}^{r_{k}}}{N_{k}}$$
where $k_{v_{w}}^{r_{j}}$ represents the degree of node $v_{w}$ on relationship $r_{j}$ and $N_{j}$ represents the total number of nodes associated with relationship $r_{j}$. If node $v_{w}$ and node $v_{h}$ are in the same subnet, then $r_{j}$ = $r_{k}$.

The greater the traffic flow on the edge, the greater the load on the nodes at both ends of the edge. In order to represent the impact of edge flow traffic on nodes in each relationship, the weight $W_{v_{w}}^{r_{i}}$ is used to represent the intensity of the effect of relationship $r_{i}$ on node $v_{w}$:(6)$$W_{v_{w}}^{r_{i}}=\sum_{v_{h}\in \Gamma_{v_{w}}}flow(<v_{w},v_{h}>)$$

If there is high traffic flow between node $v_{w}$ and its neighbor nodes, node $v_{w}$ will be greatly affected in relationship $r_{i}$.

In real life, people will comprehensively consider a variety of factors such as convenience and economy to choose the mode of public transportation. Therefore, the intensity values of multiple relationships are different. The sum of the intensity of all relationships between node $v_{w}$ and its neighboring nodes is set as:(7)$$W_{v_{w}}^{R}=\sum_{i\in R}f_{i}\cdot W_{v_{w}}^{r_{i}}$$
where $f_{i}\in \{f_{1},\ldots ,f_{i},\ldots f_{s}\}$, which represents the proportional parameter of the intensity of relationships, and different proportions have different impacts on nodes.

The initial load of a node is related to the comprehensive intensity of various relationships on the node. Therefore, in a composite network, the initial load of a node is defined as:(8)$${L_{v_{w}}^{R}(0)=\left(W_{v_{w}}^{R}\right)}^{\alpha }$$
where $L_{v_{w}}^{R}(0)$ represents the initial load of node $v_{w}$ and α is an adjustable parameter which can be used to control the initial load intensity.

(2)Node capacity

In an urban public transportation network, a larger node capacity means more passengers can be accommodated and nodes are less likely to fail. Due to cost constraints in reality, the node capacity cannot be infinitely increased. Therefore, in the model, the node capacity is proportional to its initial load.
(9)$$C_{v_{w}}=(1+c)\;L_{v_{w}}^{R}(0)$$
where $C_{v_{w}}$ represents the node capacity of node $v_{w}$.

(3)Load redistribution

When a node in an urban public transportation network fails, passengers will choose other types of transportation, and passenger flow can be described as the load flow to neighboring nodes in different ways. If failed node $v_{w}$ allocates a portion of the load to neighboring nodes $v_{h}$ based on the relationship $r_{i}$, the load will be allocated according to the proportion of the flow of node $v_{h}$ to the total flow of all neighbor nodes.
(10)$$\Delta L_{v_{w}v_{h}}^{R}(0)=\sum_{i\in R}\;L_{v_{w}}^{R}(0)\frac{W_{v_{h}}^{r_{i}}}{\sum_{vk\in \Gamma_{v_{h}}}W_{v_{k}}^{r_{i}}}$$
where $\Delta L_{v_{w}v_{h}}^{R}(0)$ represents the total redistributed load by the failed node $v_{w}$ to its non-failed neighbor node $v_{h}$ through different relationships.

As shown in Figure 2, when node $v_{5}$ fails, its load will be transmitted to node $v_{4}$ under the single relationship $r_{2}$, which means that the load is transmitted within the same subnet. The load of $v_{5}$ is also transmitted to $v_{2}$ along with relationships $r_{1}$, $r_{2}$ and $r_{3}$, transmitted to $v_{6}$ along with relationships $r_{1}$ and $r_{2}$ and transmitted to $v_{7}$ along with relationships $r_{1}$ and $r_{3}$. This means that the load is transmitted in different subnets along with various relationships. If the sum of the initial load and that received by the neighbor node is greater than the capacity, then the node will also fail (such as $v_{2}$). The node will continue to distribute the load to its non-failed neighbor nodes along with each relationship until the cascading failure process is complete.

### 2.3. Robustness Evaluation Indicator

In order to better study the robustness of the public transportation composite network, robustness evaluation indicators for public transportation composite networks are proposed in this paper. $P_{v}$ is introduced to represent the proportion of failed nodes in the entire network caused by node $v_{v}$ which is attacked.
(11)$$P_{v}=F_{v}/N-1$$
where $F_{v}$ represents the number of failed nodes and N represents the total number of nodes in the network. Obviously, the smaller $P_{v}$, the stronger the robustness of the network.

The convenience of connecting or communicating between node pairs in a network can be measured by efficiency, which depends on the shortest path. If node i and node j are connected, the shortest path between them is $d_{ij}$. Then, the connection efficiency $E_{ij}$ between two nodes is defined as follows:(12)$$E_{ij}=1/d_{ij}$$

Network efficiency E is the average connection efficiency of all node pairs in the network:(13)$$E=\frac{1}{{\left|N\right|}_{A}({\left|N\right|}_{A}-1)}\sum_{i\neq j}\frac{1}{d_{ij}}$$
where ${\left|N\right|}_{A}$ represents the number of nodes in the urban public transportation composite network.

$E_{I}$ and $E_{C}$ represent the network efficiency of urban public transportation composite networks before and after cascading failure, respectively. Therefore, changes in network efficiency $\rho \left(E\right)$∈[0, 1] can be defined as:(14)$$\rho \left(E\right)=E_{I}-E_{C}/E_{I}$$

In order to verify the robustness of the network, two attack strategies [19] are currently used: a random attack and a deliberate attack. For deliberate attacks, the node with the highest centrality S in terms of the total intensity of various relationships on the node is chosen to be attacked. S is defined as follows:(15)$$S=W_{v_{w}}^{R}/N-1$$
where $W_{v_{w}}^{R}$ represents the sum of the intensity of all relationships between node $v_{w}$ and its neighboring nodes on node $v_{w}$.

For random attack, nodes in the composite network are randomly selected for attack, and the average value is taken as the numerical result after 100 attacks in the simulation experiment.

## 3. Results

### 3.1. Experimental Data

In this paper, the data from bus and subway networks in Qingdao are used. After eliminating isolated lines and stations, the bus network contains 175 lines and 1335 stations, and the subway network contains 8 lines and 88 stations. Each station in the bus network or in the subway network is a network node. If two stations are adjacent to each other on a bus line or a subway line, the corresponding network nodes must be connected. If the distance between a bus station and a subway station is shorter than 500 m, the stations are transfer stations, and the corresponding network nodes must be connected. The edge between these nodes represents the transfer path between a bus and the subway. The bus–subway composite network is shown in Figure 3.

In Figure 3, blue line represents bus line, and red line represents subway line. There are three kinds of relationships, $r_{1}$, $r_{2}$, $r_{3}$, in the composite network, relationship $r_{1}$ indicates that both ends of the edge are bus stations, relationship $r_{2}$ indicates that both ends of the edge are subway stations and relationship $r_{3}$ indicates that the ends of the edge are a bus station and a subway station.

### 3.2. Experimental Analysis

According to the attack strategy in this paper, the Qingdao urban bus–subway composite network was attacked in a simulation. Firstly, the ratio of the three relationships was set to 1:1:1. When the initial load coefficient α is different, the change in $P_{v}$ with the capacity parameter c under a deliberate attack and a random attack is shown in Figure 4 and Figure 5.

From Figure 4 and Figure 5, we can see that in the two attacks, when α is the same, the value of $P_{v}$ decreases with the increase in c, and the network robustness also increases. This is mainly because in the case of the same relationship ratio, the larger the node capacity, the less likely it is to fail. Under a deliberate attack, the larger the value of α, the smaller the value of c required to reach the minimum value of $P_{v}$. When α = 2, the minimum value of c required to reach the minimum value of $P_{v}$ is 0.15. Under a random attack, when α = 0.5, the curve changes relatively smoothly. Unlike deliberate attacks, when α = 1.75, the minimum value of c required to reach the minimum value of $P_{v}$ is 0.1. This indicates that a deliberate attack is more destructive to the network. In order to reduce the proportion of failed nodes in the entire network caused by a deliberate attack, the initial load intensity must be expanded. Therefore, strengthening the security protection of key nodes and preventing deliberate attacks on them are the keys to improving the robustness of urban public transportation networks.

Next, α was set to 2 and the variation in $P_{v}$ with the change in capacity coefficient was analyzed under different intensity ratios of three relationships. The results under deliberate attack and random attack are shown in Figure 6 and Figure 7.

From Figure 6 and Figure 7, we can see that due to the different proportion coefficients among multiple relationships, there may be the phenomenon of a high capacity coefficient but weak network robustness. In Figure 6, when the relationship intensity ratio is 5:1:1, the value of $P_{v}$ is the lowest when c = 0.125, and when the relationship intensity ratio is 1:5:1, the minimum value of c required to reach the minimum value of $P_{v}$ is 0.3. In Figure 7, when the relationship intensity ratio is 5:1:1, the network robustness is significantly better than that of other relationship intensity ratios. When the relationship ratio of the two attack strategies is 10:1:1, the curve of $P_{v}$ is better than that when the ratios are 1:1:10 and 1:10:1. Therefore, under the two attacks, we find that when the proportions of $r_{2}$ and $r_{3}$ are the same, and the proportion of $r_{1}$ is higher, the robustness of the network is stronger. This is mainly because there are far more bus stations than subway stations. When the relationship intensity value of bus stations is large, the network robustness is relatively strong.

In order to study the impact of relationship $r_{1}$ on network robustness, the ratio of $r_{2}$ to $r_{3}$ is fixed as 1:1 and the proportion of the relationship intensity of $r_{1}$ is changed; the values of $P_{v}$ are shown in Figure 8 and Figure 9.

From Figure 8 and Figure 9, we can see that when $r_{2}:r_{3}$ = 1:1, as the value of $r_{1}$ increases and the value of c required to reach the minimum value of $P_{v}$ also increases. As shown in Figure 8, under deliberate attack, when $r_{1}:r_{2}:r_{3}$ = 5:1:1, the value of $P_{v}$ is smallest when c = 0.125. As shown in Figure 9, under random attack, when $r_{1}:r_{2}:r_{3}$ = 5:1:1, the value of $P_{v}$ is smallest when c = 0.225. This indicates that both random and deliberate attacks have a consistent impact on the proportion of relationships in composite networks.

In order to accurately gauge the contributions and advantages of the improved cascading failure model in this paper, the proposed model is compared with a flow-based cascading failure model [20] for urban public transportation networks to verify its advantages. Under random attack and deliberate attack, when the relationship proportions and parameter c are the same, the minimum values of $P_{v}$ and $\rho \left(E\right)$ of the two cascading failure models for urban public transportation networks are shown in Table 1.

It can be seen from Table 1 that the changes in network efficiency and the proportion of failed nodes of the proposed cascading failure model are smaller than those of the flow-based cascading failure model, especially for cascading failures caused by deliberate attacks on important nodes. The proportion of failed nodes is significant better than the flow-based cascading failure model. This indicates that the proposed cascade failure model in this paper can effectively enhance the robustness of urban public transportation composite networks.

## 4. Discussion

An urban public transportation network is a complex and dynamic high-order network. The analysis of cascading failures in transportation networks can better explain the mechanism of traffic congestion and the laws of its propagation. However, existing studies have mostly focused on the static network topology, node characteristics and load changes of transportation networks which are constructed by a single public transportation vehicle; they neglect the fact that urban public transportation networks are actually composed of multiple modes of transportation.

In this paper, high-order complex network theory is used to model the urban public transport system as a multi-relationship complex network, and an urban bus–subway composite network is constructed. The composite network is more in line with the real situation of urban transportation networks, and cascading failure research in the composite network can more realistically reflect the robustness of urban public transportation networks.

Previous public transportation network studies focused on cascading failures of urban rail transit networks or bus transit networks while ignoring their coupled performance against cascading failures. In urban public traffic composite networks, cascading failures of multi-modal public transit networks exhibit complex coupling dynamics caused by the intertwining and coupling effect of two types of failure load dynamic redistributions. Therefore, this paper provides a novel perspective to analyze the urban public traffic network robustness based on a high-order complex network. Some researchers adopted similar research approaches; Yildirimoglu et al. [21] studied the coupling relationship between multiple transportation modes. However, they only used a multi-layer network model to connect different traffic patterns and did not delve deeply into the relationships between different layers. Ameli et al. [22] proposed a multimodal composite traffic network model and analyzed the characteristics of multimodal traffic networks. Their model can automatically generate the topology structure of a multimodal traffic network. However, an urban transportation system is a dynamic system, and cannot be accurately described by a static model. At the same time, the application of multimodal urban public transportation is mostly limited to the application of transportation multi-source data, without truly utilizing the topology structure and the relationship between different transportation modes to analyze real transportation problems. For example, Zhang et al. [23] used connections between different transportation modes and considered various combination travel modes such as bus–taxi, bus–subway and taxi–subway, but they did not conduct a quantitative analysis of the specific impact of different travel modes on the cascading failure problem.

Aiming at these problems in the urban public transportation composite network research field, a cascading failure model for urban public traffic networks is proposed in this paper. Through simulation experiments on the constructed urban bus–subway composite network, it has been proven that when the proportion of relationships is different and as the capacity coefficient increases, the robustness of the urban public traffic network may not necessarily increase. Therefore, the proportional coefficient between the intensity of multiple relationships directly affects the robustness of the urban public traffic network. When both ends of the edge in the transportation composite network are subway stations or both ends are bus subway stations, the ratio of relationship intensity between the bus stations can be appropriately increased, which can effectively enhance network robustness. Our work can provide decision support for urban public transportation system operation management.

The simulation analysis results of this paper have practical significance for the safety of urban public transportation networks. According to the analysis of the experimental results, it was found that the initial load coefficient can have a certain impact on the robustness of urban public transportation networks, and the larger the capacity coefficient, the stronger the network robustness. However, in actual public transportation networks, these two coefficients are difficult to quantify. Therefore, from a macro perspective, to enhance network robustness, it is necessary to strengthen the protection of important nodes and develop passenger evacuation plans in advance for each station to prevent trapping passengers in dangerous situations and congestion at other stations or roads caused by irrational passenger flow allocation methods. At the same time, in the construction of public transportation networks, it is necessary to strengthen the construction of rail transit to alleviate the pressure on public transportation. In addition, by comparing the relationship intensity ratios of different types of public transportation, it can be seen that the bus network has a greater impact on the robustness of the entire urban public transportation composite network. However, in practice, increasing the capacity of bus stations is constrained by many factors; it is difficult to make the capacity of bus stations sufficiently large. Therefore, it is necessary to organically connect urban rail transportation and urban buses and coordinate the proportional relationship between the two. When the relationship intensity ratio between subway stations on both ends of the connecting edge and bus–subway stations on both ends is the same, the relationship intensity ratio between bus stations can be appropriately increased, which can effectively enhance network robustness. Only in this way we can minimize engineering costs while ensuring the robustness of the entire network in urban transportation network planning and enable the construction and operation of the entire urban public transportation network to enter a virtuous cycle.

## 5. Conclusions

Based on a multi-subnet composite complex network, in this paper, an urban public traffic composite network is constructed and a new cascading failure model of urban composite networks is proposed. This model considers the impact of different types of public transportation on the robustness of urban public transportation networks, and the load redistribution mechanism is more in line with the actual situation of urban public transportation networks. Additionally, through comparative experiments, it has been proven that the proposed model has significant advantages in terms of network efficiency and the proportion of network failure nodes compared to the existing cascaded failure models of urban public transportation composite networks. Through the simulation of an actual urban bus–subway network, it is proven that the proportional coefficient between multiple relationship affects the network robustness. This means that when more passengers choose to travel by bus than by subway, the robustness of the urban public traffic composite network is stronger. In order to better ensure the robustness of urban public transportation networks, the focus of future research should be on finding the optimal proportion of the relationship intensity, and some optimization measures can be used as new control parameters of the proposed cascading failure model.

## Figures and Tables

**Figure 1 entropy-25-01377-f001:**
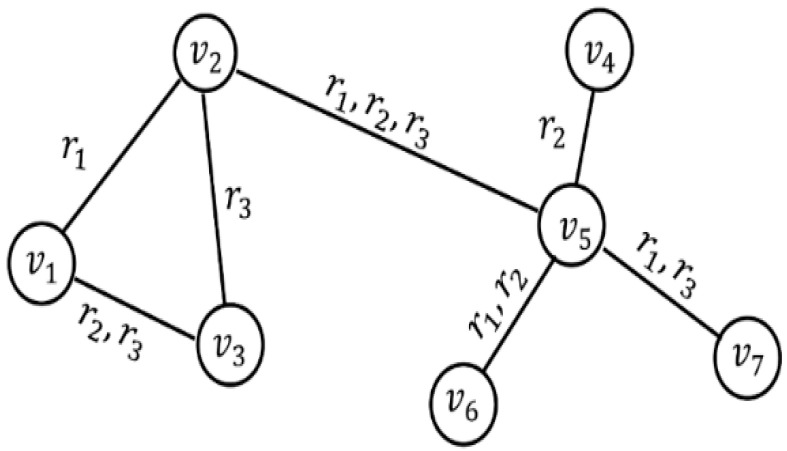
Example of a multi-subnet composite complex network.

**Figure 2 entropy-25-01377-f002:**
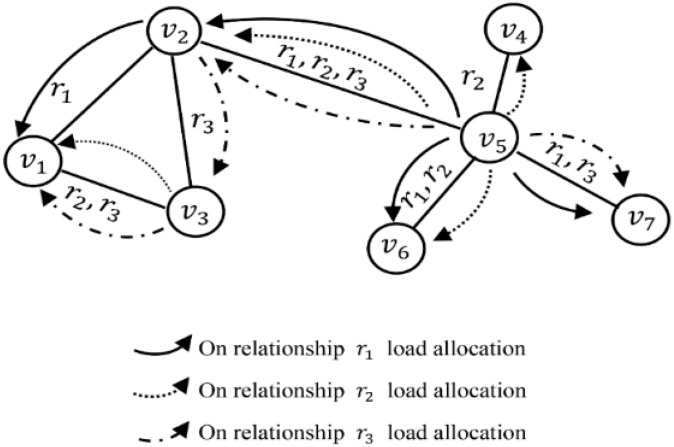
Schematic diagram of load redistribution in a composite network.

**Figure 3 entropy-25-01377-f003:**
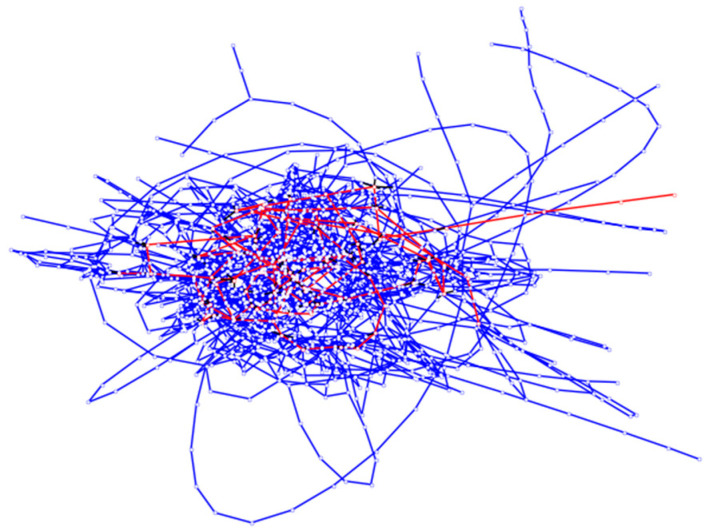
Topology of Qingdao’s bus–subway composite network.

**Figure 4 entropy-25-01377-f004:**
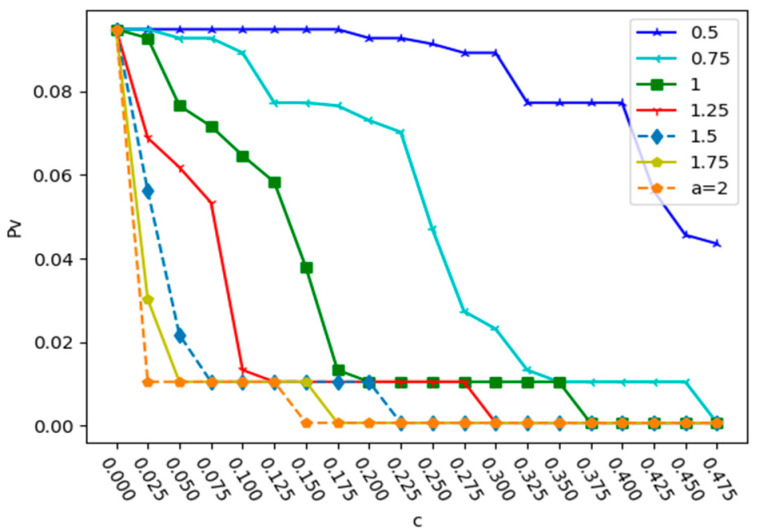
Changes in proportion $P_{v}$ with the capacity parameter c with different α under deliberate attack.

**Figure 5 entropy-25-01377-f005:**
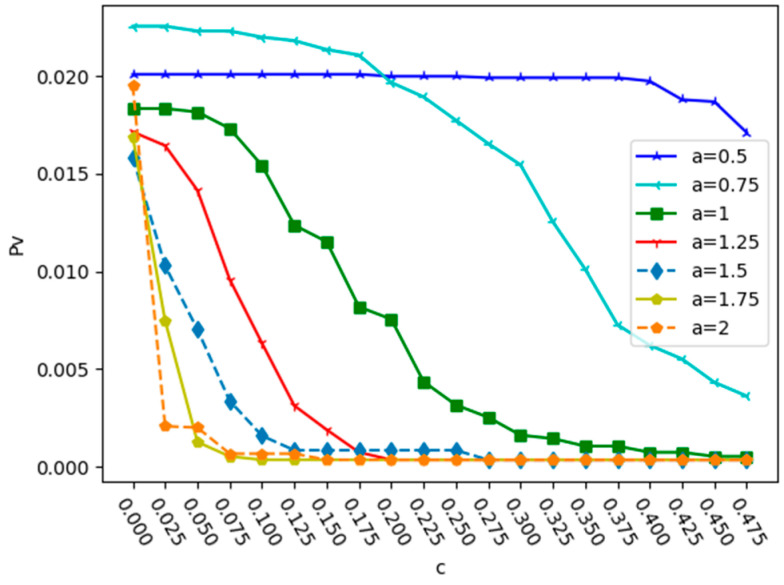
Changes in proportion $P_{v}$ with the capacity parameter c with different α under random attack.

**Figure 6 entropy-25-01377-f006:**
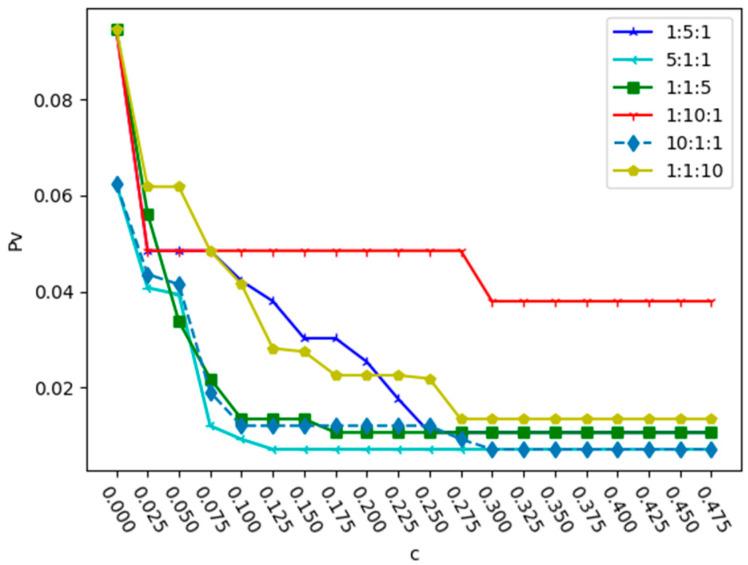
Changes in proportion $P_{v}$ with the capacity parameter c in different relationship proportions under deliberate attack.

**Figure 7 entropy-25-01377-f007:**
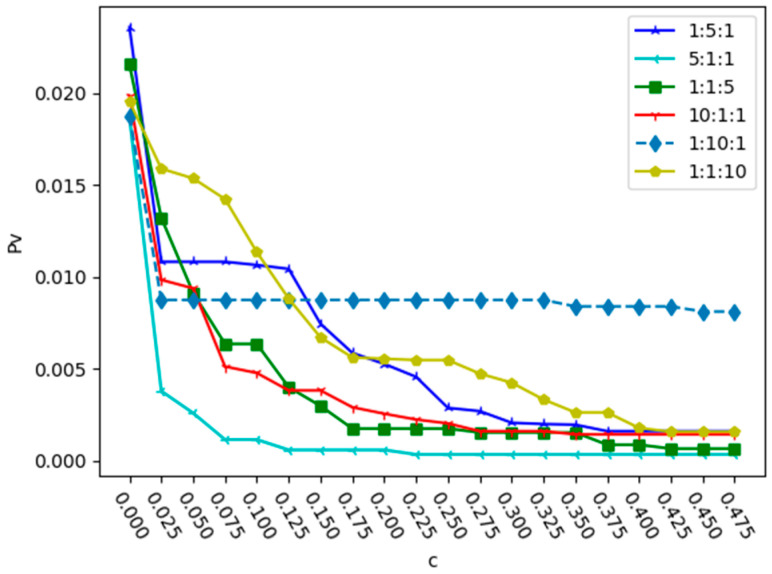
Changes in proportion $P_{v}$ with the capacity parameter c in different relationship proportions under random attack.

**Figure 8 entropy-25-01377-f008:**
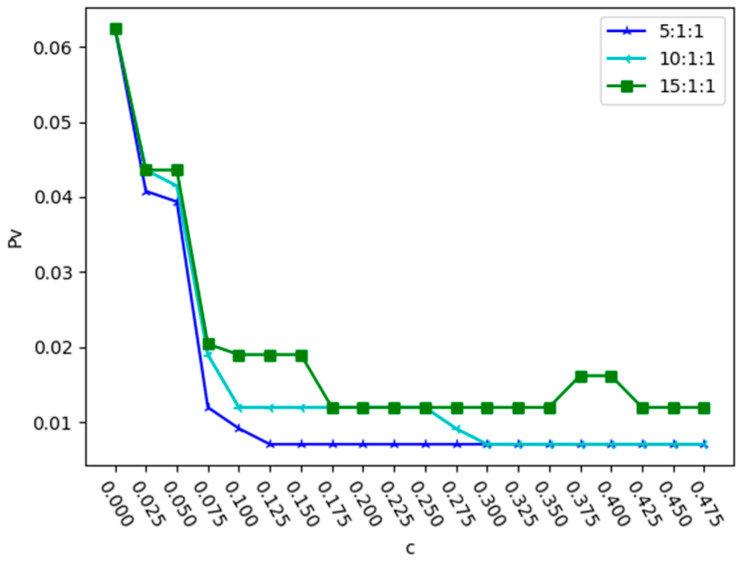
Changes in proportion $P_{v}$ with the value of $r_{1}$ when $r_{2}:r_{3}$ = 1:1 under deliberate attack.

**Figure 9 entropy-25-01377-f009:**
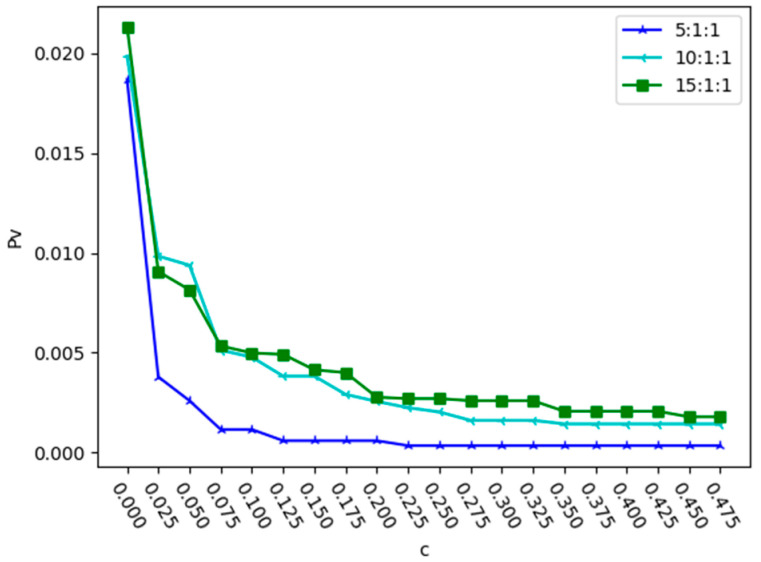
Changes in proportion $P_{v}$ with the value of $r_{1}$ when $r_{2}:r_{3}$ = 1:1 under random attack.

**Table 1 entropy-25-01377-t001:** Comparison results of two cascading failure models for urban public transportation networks.

Attack Mode	Random Attack		Deliberate Attack	
Evaluation Indicator	$P_{v}$	$\rho \left(E\right)$	$P_{v}$	$\rho \left(E\right)$
The proposed cascading failure model	0.1	0.338	0.125	0.431
Flow-based cascading failure model	0.16	0.403	0.24	0.527

## Data Availability

Not applicable.

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
