# Peer review of "Robustness Analysis of an Urban Public Traffic Network Based on a Multi-Subnet Composite Complex Network Model"

_entropy, 2023, doi:10.3390/e25101377_

Round 1
Reviewer 1 Report
This article employs the Multi-subnet Composited Complex Network model to establish a framework for analyzing cascading failures in urban transportation systems and conducts an investigation of real-world urban transportation networks. In a comprehensive assessment, this paper offers certain insightful implications. My concerns regarding this article are primarily as follows:
The article conducts a thorough analysis of an actual urban transportation network and presents a series of simulated analytical outcomes. However, a deficiency lies in the lack of a comprehensive appraisal and analysis of the efficacy of these results.
Notably, within the context of practical transportation network scenarios, the article does not adequately contrast the merits and demerits of its proposed model with those of existing alternative models. A meticulous examination in this aspect is imperative for accurately gauging the contributions and advantages of the proposed model.
A typo in equation (3): one of Vk should be v_k(subscript)
Author Response
Dear Editor,
Thank you very much for your letter and for the comments by the reviewers. These comments are very valuable and helpful for my paper.
I appreciate the careful, constructive, and generally favorable reviews given to our paper by the reviewers.
I believe that I have adequately addressed all the excellent advices and questions raised by reviewers. Furthermore, I checked the manuscript and made sure the submitted manuscript is correct.
Response to the comments of reviewer 1:
Point 1: The article conducts a thorough analysis of an actual urban transportation network and presents a series of simulated analytical outcomes. However, a deficiency lies in the lack of a comprehensive appraisal and analysis of the efficacy of these results. 

Response 1: Thanks for the comment. I had added a comprehensive appraisal and analysis of the efficacy of these results from practical significance for the safety of urban public transportation networks in Discussions Section.
Point 2: Notably, within the context of practical transportation network scenarios, the article does not adequately contrast the merits and demerits of its proposed model with those of existing alternative models. A meticulous examination in this aspect is imperative for accurately gauging the contributions and advantages of the proposed model. 

Response 2: Thanks for the comment. I had contrasted the merits of its proposed model with an existing alternative model at the end of Section 3.2.
Point 3: A typo in equation (3): one of Vk should be v_k(subscript). 

Response 3: Thanks for the comment. I had revised the error.
Reviewer 2 Report
In this paper, the authors introduced composited networks, by collapsing multiple coupled network layers into one, and collecting information from the links in the original layers as variable length labels on the composited network. By making these labels real variables that represent the maximum flows possible, the authors could also test the robustness of the composited network against random as well as targetted attacks as a function of the total flow. They applied this modelling framework to the bus and subway network, where the nodes are bus stations or joint bus/subway stations, and the links tell us how many commuters can be carried by buses or trains per unit time. The authors then performed random/targetted attacks on the network. After a selected node is attacked, the authors follow the cascading failure as it propagates upstream of the node attacked. A node fails if the incident flow is greater than its capacity. The authors determine the proportion of failed nodes, and use this quantity as a measure of the robustness of the network.
As pointed out by the authors, the novelty of this paper lies in the use of composited networks, where links carry more than one weight, and their tests of the robustness of a real-world public transportation network, subjected to random/targetted attacks, as a function of the total capacity. As expected, the public transportation network becomes more robust as the total capacity is increased. Based on the novelty of the network model, and the test of robustness as capacity is varied, I would recommend for this paper to be published.
However, I have several concerns that I would like the authors to react to, before publication. These are:
(1) It seems that it is not necessary to have weights of the form (r1, r2, r3) for the links, since r1 is the capacity for buses, and r2 is the capacity for trains. If we adopt the vector-valued weight (r1, r2) to each link, we then can tell whether two nodes are linked by buses, by trains, or by both.
(2) In a city, travel demands do not originate at or end at bus stops or train stations. Instead, they originate at homes, offices, malls, and other locations of interest. Therefore, the bus-subway composited network would miss the other modes of transportation, which are private cars, taxis, and walking. If a city is walkable, minor disruptions may not even be felt, because commuters can always choose to walk slowly to another part of the bus-train network, or even to the destination. When these are not known, I worry that it is premature to conclude on the robustness of the bus-train composited network.
Here, I am not suggesting the authors redo their entire analysis. However, I would urge the authors to estimate the capacities contributed by walking, taxis, and private cars, since a disruption in either the bus or subway network can be compensated by walking or taking cabs.
(3) There are several subtle differences between the results for random attacks and those for targetted attacks. The descriptions of many of these are superficial, and must be improved before publication.
(4) The authors called Section 4 Discussion. However, the content in this section is largely a continuation of the literature survey contained in Section 1. Therefore, these should be moved to Section 1.
A proper discussion should be Discussion should be about the results obtained, in particular how they can be understood. I would also like to see the authors discuss how their findings can be applied to public transportation planning.
(5) The Conclusions is where we would summarise the findings of our paper. However, the conclusions section found in this paper is short, and contains only vague sentences. This must be improved.
Another reason the paper needs to be revised is the language used in the paper. There are not many obvious grammatical errors in the paper, but one can find many sentences whose meanings are not clear.
For example, in line 231, we find "In this paper, the data of bus and subway networks in Qingdao urban is used". 'Urban' is an adjective, so it should not follow after the place name 'Qingdao'.
In line 241-242, we find "..., relationships ?3 represents the two stations are one bus station and one subway station", which is grammatically problematic.
There are other problems elsewhere, so I would recommend the authors engage an editor they trust, to identify these language problems, and write the paper in a way that most readers would be able to understand.
Author Response
Dear Editor,
Thank you very much for your letter and for the comments by the reviewers. These comments are very valuable and helpful for my paper.
I appreciate the careful, constructive, and generally favorable reviews given to our paper by the reviewers.
I believe that I have adequately addressed all the excellent advices and questions raised by reviewers. Furthermore, I checked the manuscript and made sure the submitted manuscript is correct.
Response to the comments of reviewer 2:
Point 1: It seems that it is not necessary to have weights of the form (r1, r2, r3) for the links, since r1 is the capacity for buses, and r2 is the capacity for trains. If we adopt the vector-valued weight (r1, r2) to each link, we then can tell whether two nodes are linked by buses, by trains, or by both. 

Response 1: Thanks for the comment. The vector-valued weight (r1, r2) to each link is completely right, and in Multi-subnet Composited Complex Network Model, multiple relationships on edges are represented in the form of vectors. We also use this form in the specific simulation programming process.
Point 2: In a city, travel demands do not originate at or end at bus stops or train stations. Instead, they originate at homes, offices, malls, and other locations of interest. Therefore, the bus-subway composited network would miss the other modes of transportation, which are private cars, taxis, and walking. If a city is walkable, minor disruptions may not even be felt, because commuters can always choose to walk slowly to another part of the bus-train network, or even to the destination. When these are not known, I worry that it is premature to conclude on the robustness of the bus-train composited network. 
Here, I am not suggesting the authors redo their entire analysis. However, I would urge the authors to estimate the capacities contributed by walking, taxis, and private cars, since a disruption in either the bus or subway network can be compensated by walking or taking cabs.
Response 2: Thanks for the comment. As you said, there are many modes of transportation in the urban transportation system, such as, walking, taxis, and private cars. But the data on these modes of travel, especially walking, is very difficult to quantify and obtain. This paper mainly studies the urban public transportation network, so bus network and subway network are chosen. These are the two networks commonly used in research in this field. In the future research, we will also obtain more data on other modes of travel to further improve our research.
Point 3: There are several subtle differences between the results for random attacks and those for targetted attacks. The descriptions of many of these are superficial, and must be improved before publication. 

Response 3: Thanks for the comment. I had improved the analysis of the results for random attacks and those for targetted attacks.
Point 4: The authors called Section 4 Discussion. However, the content in this section is largely a continuation of the literature survey contained in Section 1. Therefore, these should be moved to Section 1. A proper discussion should be Discussion should be about the results obtained, in particular how they can be understood. I would also like to see the authors discuss how their findings can be applied to public transportation planning.
Response 4: Thanks for the comment. I had moved the literature survey from Section 4 to Section 1. I also added discussion about the results and discussed how my findings can be applied to public transportation planning in the end of Discussion Section.
Point 5: The Conclusions is where we would summarise the findings of our paper. However, the conclusions section found in this paper is short, and contains only vague sentences. This must be improved.
Response 5: Thanks for the comment. I had improved the Conclusions Section.
Point 6: Another reason the paper needs to be revised is the language used in the paper. There are not many obvious grammatical errors in the paper, but one can find many sentences whose meanings are not clear.
For example, in line 231, we find "In this paper, the data of bus and subway networks in Qingdao urban is used". 'Urban' is an adjective, so it should not follow after the place name 'Qingdao'.
In line 241-242, we find "..., relationships ??3 represents the two stations are one bus station and one subway station", which is grammatically problematic.
There are other problems elsewhere, so I would recommend the authors engage an editor they trust, to identify these language problems, and write the paper in a way that most readers would be able to understand.
Response 6: Thanks for the comment. I had revised those mistakes you pointed out, and improved the English of the paper carefully.
Reviewer 3 Report
In this paper, the authors have analysed urban public transportation networks using complex network theory. Following are my comments: 1. The overall writing of the paper needs to be improved. For example, the authors have used v_v to represent a node. This is very confusing as the symbol v is used twice here. I am also not clear what is the difference between 2. Figure 3 is not conveying any useful information. 3. My major concern is about the experimental evaluations. The quantitative results are limited. The authors have not provided any comparison with other methods.The writing of the paper needs to be improved.
Author Response
Dear Editor,
Thank you very much for your letter and for the comments by the reviewers. These comments are very valuable and helpful for my paper.
I appreciate the careful, constructive, and generally favorable reviews given to our paper by the reviewers.
I believe that I have adequately addressed all the excellent advices and questions raised by reviewers. Furthermore, I checked the manuscript and made sure the submitted manuscript is correct.
Response to the comments of reviewer 3:
Point 1: The overall writing of the paper needs to be improved. For example, the authors have used v_v to represent a node. This is very confusing as the symbol v is used twice here. I am also not clear what is the difference between. 

Response 1: Thanks for the comment. I had revised these symbols.
Point 2: Figure 3 is not conveying any useful information. 

Response 2: Thanks for the comment. Figure 3 mainly represents the proportion of bus and subway lines in the composited network through different colours.
Point 3: My major concern is about the experimental evaluations. The quantitative results are limited. The authors have not provided any comparison with other methods.
Response 3: Thanks for the comment. I had added the quantitative comparison with existing model at the end of Section 3.2.
Point 4: The writing of the paper needs to be improved.
Response 4: Thanks for the comment. I had improved the English of the paper carefully.
Round 2
Reviewer 2 Report
I am satisfied with the responses by the authors to my comments, and the revisions they have made.
Reviewer 3 Report
I have no further comments.
I have no further comments.